

# A multiomics comparison between endometrial cancer and serous ovarian cancer

Hui Zhong[1], Huiyu Chen[1], Huahong Qiu[1], Chen Huang[2] and Zhihui Wu[1]

[1] Department of Clinical Laboratory, Fujian Provincial Maternity and Children's Hospital, Affiliated hospital of Fujian Medical University, Fuzhou, Fujian, China
[2] Lester and Sue Smith Breast Center, Baylor College of Medicine, Houston, TX, United States of America

## ABSTRACT

**Background.** Endometrial carcinoma (EC) and serous ovarian carcinoma (OvCa) are both among the common cancer types in women. EC can be divided into two subtypes, endometroid EC and serous-like EC, with distinct histological characterizations and molecular phenotypes. There is an increasing awareness that serous-like EC resembles serous OvCa in genetic landscape, but a clear relationship between them is still lacking.
**Methods.** Here, we took advantage of the large-scale molecular profiling of The Cancer Genome Atlas(TCGA) to compare the two EC subtypes and serous OvCa. We used bioinformatics data analytic methods to systematically examine the somatic mutation (SM) and copy number alteration (SCNA), gene expression, pathway activities, survival gene signatures and immune infiltration. Based on these quantifiable molecular characterizations, we asked whether serous-like EC should be grouped more closely to serous OvCa, based on the context of being serous-like; or if should be grouped more closely to endometroid EC, based on the same organ origin.
**Results.** We found that although serous-like EC and serous OvCa share some common genotypes, including mutation and copy number alteration, they differ in molecular phenotypes such as gene expression and signaling pathway activity. Moreover, no shared prognostic gene signature was found, indicating that they use unique genes governing tumor progression. Finally, although the endometrioid EC and serous OvCa are both highly immune infiltrated, the immune cell composition in serous OvCa is mostly immune suppressive, whereas endometrioid EC has a higher level of cytotoxic immune cells. Overall, our genetic aberration and molecular phenotype characterizations indicated that serous-like EC and serous OvCa cannot be simply treated as a simple "serous" cancer type. In particular, additional attention should be paid to their unique gene activities and tumor microenvironments for novel targeted therapy development.

Corresponding author
Zhihui Wu, 13799386698@139.com

## INTRODUCTION

Endometrial carcinoma (EC) and ovarian carcinoma (OvCa) are two common female cancers, accounting for 4th and 5th-leading causes of cancer death among women in the United States (*Siegel, Miller & Jemal, 2019*). EC can be divided into two subgroups, a

type I endometrioid tumors and a type II serous-like tumor (*Getz et al., 2013*). Compared to type I EC, type II serous-like EC was characterized with a more advanced stage and worse outcome. As for ovarian cancers, high grade OvCa serous tumors account for the most cancer death (*Matulonis et al., 2016*). Previous studies have identified similar genetic aberrations among serous-like EC and serous OvCa (*Getz et al., 2013*). For example, both serous-like EC and serous OvCa have frequent *TP53* mutation, whereas type I endometroid EC does not. Also, serous-like EC and serous OvCa are both featured with chromosome instability and copy number alteration (CNA), compared to very few CNA events in type I endometroid EC. These findings suggest that they might be caused by similar oncogenic drivers and more importantly, share common molecular mechanisms for tumor progression.

In recently years, an idea in understanding and targeting cancer for treatment has been brought up that cancers need to be classified by genetic similarity rather than tissue or organ origins (*Heim et al., 0000*; *Margolin et al., 2014*). The rationale of such classification is that, cancer types sharing similar cancer genetic drivers and progression mechanisms are more likely to be targeted using common drugs, regardless of their tissue- or organ-origin (*Aggarwal, 2010*). Following this rule, there is a possibility that type II serous-like EC might be classified together with serous OvCa, instead of being classified with type I endometroid cancer. The current treatment strategies also reflect such similarities: both serous-like EC and serous OvCa are commonly treated with platinum- or taxane-based chemotherapies, although the responsiveness varies (*Moxley & McMeekin, 2010*; *Brasseur, Gévry & Asselin, 2017*; *Cortez et al., 2018*). In comparison, type I endometroid cancer is more frequently treated with adjuvant radiotherapy (*Hopkins Hospital et al., 2017*). Also, there is not a single targeted therapy that works well for serous-like EC and serous OvCa, but type I endometroid EC patients might be treated with immunotherapy (*Piulats & Matias-Guiu, 2016*). This is because some endometroid EC tumors are featured with microsatellite instability (MSI) and genomic hypermutation, which can be translated into neoantigens to attract cytotoxic immune infiltration. Therefore, it is extremely useful to have a deeper understanding of serous-like EC and OvCa. In particular, whether these two cancer types are similar enough to be categorized together (bypassing the different tissue origins) and be treated by common anti-tumor drug target identification needs to be clearly defined.

The Cancer Genome Atlas (TCGA) has generated large-scale omics data for more than 32 cancer types (*Wang, Jensen & Zenklusen, 2016*). The high-throughput profiling effort has led to unprecedent understanding of somatic mutation, copy number changes, gene expression and other molecular phenotypes of each tumor type. Moreover, the TCGA data also provide a unique chance to compare the genetic aberrations across different cancer types and even allow for pan-cancer studies. For instance, a recent study used TCGA large-scale data to compare gynecologic cancers and breast cancer (*Berger et al., 2018*). By unsupervised analyses, this study revealed that a subset of EC samples, particularly those belonging to serous-like EC, can be clustered together with OvCa in genetic aberration and gene expression. One potential drawback for this type of "pan-cancer" analyses, however, is that the involvement of too many cancer types might compromise the ability to distinguish some subtle yet significant differences across some specific cancer types. To

**Table 1** **The sample size (N) for each omics data type.** Numbers in parentheses indicate the overlap of each omics data size with the clinical data.

| Cancer types<br>Data type | Endometroid EC | Serous-like EC | Serous OvCa |
|---|---|---|---|
| Clinical data | $N = 411$ | $N = 115$ | $N = 587$ |
| Data type RNA expression | $N = 409$ (409) | $N = 114$ (114) | $N = 308$ (303) |
| Somatic copy number alternation | $N = 391$ (391) | $N = 110$ (110) | $N = 436$ (436) |
| Somatic mutation | $N = 293$ (293) | $N = 65$ (52) | $N = 549$ (538) |

our best knowledge, there has not been a study focusing on a "side-by-side" comparison of the two types of EC and OvCa, which share very close cancer tissue origin and tissue development (*Mullen & Behringer, 2014*; *Hoadley et al., 2018a*).

Here, we take advantage of the large-scale multi-omics data generated from TCGA and perform a comprehensive comparison of molecular profiles among endometroid EC ($N = 411$), serous-like EC ($N = 115$) and serous OvCa ($N = 587$) (see Table 1). We found that although serous-like EC and serous OvCa share some common genetic drivers, they differ in multiple biological processes, including pathway activity, prognostic gene signature and immune cell infiltration. We conclude that serous-like EC and serous OvCa use different molecular mechanisms to progress and therefore, targeted therapies based on gene and pathway functions should be uniquely adapted to counter each of them.

## MATERIALS & METHODS

### Data download

The TCGA data were downloaded by the R package "TCGAbiolinks"(*Colaprico et al., 2016*), except that the pre-normalized RNA-seq (RSEM) data were downloaded from Broad GDAC Firehose (https://gdac.broadinstitute.org/). All the data were formatted as data matrices in R. The RNA-seq data were combined and re-normalized together using upper-quantile normalization (*Bullard et al., 2010*). For missing values of RNA expression ( < 5% of all the RNA expression matrix entries), K-nearest neighbor (KNN) method (implemented in the "DMwR" R package version 0.4.1) was used to impute them.

### Somatic copy number alternation (SCNA) and somatic mutation (SM)

For SCNA analysis, all the segment-level log2 ratios were plotted out as a heatmap (Fig. 1A) to reflect the genome-wide SCNA profiles for these three cancer types. To quantify the chromosome instability, we used a previously published method (*Vasaikar et al., 2019*) that sums up the absolute segment-level log2 ratios for all the segments located in the same chromosome arm, while the segment lengths were weighted during the summation. The arm-level SCNAs were used to reflect the chromosome instability for these three cancer types.

For the SM analysis, the genome-wide mutation burden was inferred by summing all the recorded SM events (i.e., the sum of all the mutation sites across all genes) within the TCGA Mutation Annotation Format (MAF) files, regardless the location of the SM events. This was essentially the same way used by the R package "maftools" (*Mayakonda et al.,*
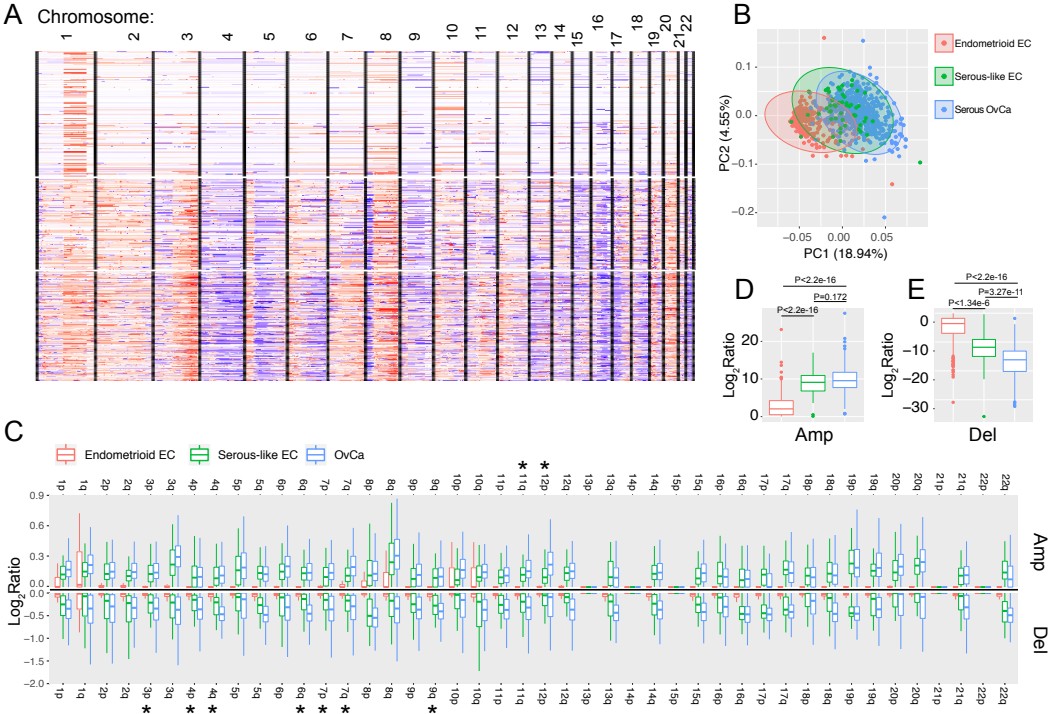

**Figure 1  The somatic copy number alternation (SCNA) of the three cancer types.** (A) Heatmap showing the SCNA landscape. Each row corresponds to one TCGA sample while each column corresponds to one chromosome. (B) PCA plot displaying the quantifications of all chromosome instabilities (i.e., summarized genome-wide total SCNAs) for these three cancer types. (C) Box plot comparing the arm-level amplification and deletion for all the 22 autosomes. Note that for every arm, the SCNA strength is weaker in endometroid EC compared to serous-like EC or serous OvCa. The star marks (*) denote the arms whose SCNAs are stronger in serous OvCa compared to serous-like EC (adjusted $p$ value < 0.05, Student's $t$-test). (D–E) The comparison of summarized genome-wide SCNA strengths (amplification and deletion, respectively) across the three cancer types. $P$ values were resulted from Student's $t$-test.

*2018*), and has been similarly adopted in previous studies (*Rooney et al., 2015*; *Li et al., 2016*). The most frequently mutated genes were also summarized from the MAF files.

To check the major genetic events (i.e., "copy-number driven" vs. "mutation-driven") driving these three cancer types (*Ciriello et al., 2013*), we calculated the copy number alteration rates and mutation rates for the most significant oncogenic and tumor suppressor genes (~200) identified in the previous pan-cancer study (*Ciriello et al., 2013*). One-side Kolmogorov–Smirnov test (i.e., "greater" vs. "less") was performed to determine the genetic event dominating the given cancer type.

## RNA expression and Pathway activity inference

To investigate the gene expressional similarity between any two of the three cancer types, we used sample-wise Spearman's correlation to compute the correlation coefficents between any sample pair across different tumor types. We used the tissue-specific (i.e., uterus and ovary) genes that were summarized previously (*Liu et al., 2008*) to detect the tissue specificity of these cancer types (Table S1). Single sample GSEA (ssGSEA) algorithm (*Barbie*

*et al., 2009*) implemented in the R package "GSVA" (version 3.6) (*Hanzelmann, Castelo & Guinney, 2013*) was used to summarize the overall tissue-specific gene expression and calculate the pathway activity for the hallmark pathways (*Liberzon et al., 2015*). One-way ANOVA was used to select variable pathways across the three cancer types (adjusted $p$ value < 0.01), and the Benjamini and Hochberg (BH) method (*Benjamini & Hochberg, 1995*) was used to adjust the multi-testing $p$ values.

## Survival analysis

We used univariate Cox proportional-hazards (PH) regression (the "coxph" function in the R package "survival", version 2.44) (*Terry & Therneau, 2019*) to quantify the contribution of each gene expression to survival outcome for the three cancer types. To infer the pathways that were enriched with prognostic genes, we extract the Cox PH regression coefficients from the models and rank them from low (worst prognostic) to high (best prognostic) and used GSEA (*Subramanian et al., 2005*) implemented in WebGestalt (*Liao et al., 2019*) to identify the KEGG pathways enriched with bad prognostic or good prognostic genes (FDR < 0.01). We used log-rank test to determine the significance of association between a given gene and survival outcome. In this survival test, the samples were dichotomized to "low" and "high" expression groups based on the median expression of the gene.

## Immune infiltration and composition inference

We used ESTIMATE (*Yoshihara et al., 2013*) to infer the overall immune and stromal infiltration and CIBERSORT (*Newman et al., 2015*) to infer the detailed immune composition for the three cancer types. Both of these two tools utilize the normalized RNA expression. To simplify immune cell analysis and provide a more straightforward results, we employed a similar strategy as the one used previously to combine all the NK cells and macrophages from different NK cell subtypes and macrophage subtypes (*Thorsson et al., 2018*). The prognostic value of CD8 T cells was inferred using log-rank test as described above.

## Statistics

The statistical approaches were described partially in the text and the Method sections above. For any tests that have not been covered, the between-group difference was tested by student-t test, and variability among three groups was tested by one-way ANOVA. The pathway enrichment was tested by GSEA imputation. The survival association was tested by log-rank test. A $p$ value less than 0.01 (or adjusted $p$ value in multi-testing) was considered as statistically significant. All the data processing and statistical analysis were performed under the R computing environment (R 3.6.0).

## RESULTS

### Copy number alteration

We first examined the somatic copy number alteration (SCNA), one of the major genetic events driving tumorigenesis. Consistent with previous report (*Getz et al., 2013*; *Berger et al., 2018*), we identified very similar SCNA patterns for EC and OvCa. While type I

endometroid EC has few SCNA events, serous-like EC and OvCa are both featured by high SCNA profiles. They both have obvious 3q, 5p, 8q and 20 p and 20q gain and 4q, 5q and 16q loss (Fig. 1A). Type I endometroid EC samples, on the other hand, are only characterized by 1q gain and no obvious arm-level loss. We further quantified the chromosome instability from all the arm level SCNAs (method) and found that serous-like EC and serous OvCa are more similar to each other than endometroid EC (Fig. 1B).

Furthermore, we systematically compared copy number gain and loss across all the autosomes and found that for all the quantifiable chromosome arms, both gain and loss are significantly stronger in the two serous cancer types than endometroid EC (adjusted $p$ value $< 0.05$, Student's $t$-test Fig. 1C). Interestingly, while the two serous cancer type have similar levels of copy number gain, OvCa has even stronger copy number loss (Figs. 1C–1E) than serous-like EC. Representative chromosome arms include chromosome 4p and 4q (adjusted $p$ value $< 0.05$, Student's $t$-test). Chromosome 4 is enriched with tumor suppressor genes and SCNA events related with this chromosome have been linked to several types of cancer (*Wang et al., 1999*; *Shivapurkar et al., 1999*; *Singh et al., 2007*). Indeed, we identified multiple tumor suppressor genes encoded in chromosome 4, including *CASP3*, *FBXW7* and *TET2* that were not only show differential loss comparing serous EC to endometroid EC, but also show additional loss in serous OvCa (Fig. S1).

## Somatic mutation

Next, we examined the somatic mutation (SM) profile in these three cancer types and also the SM genes. The endometroid EC displayed a higher genome-wide somatic mutation burden than serous-like EC and serous OvCa (Fig. 2A). This was expected, due to that some of endometroid EC tumors have microsatellite instability, which causes large amount of somatic mutation (*Getz et al., 2013*). As for gene-level SM, the frequently muted genes for serous-like EC and OvCa both include *TP53*, *MUC16*, *FLG* and *AHNAK*. Notably, except *TP53*, almost all other mutated genes have frequencies less than 25%, suggesting that these two cancer types are belong to "copy number-driven" cancer (*Ciriello et al., 2013*). In comparison, the top mutated genes in endometroid cancers have much higher frequencies, including *PTEN*, *ARID1A*, *PIK3CA*, *PIK3R1*, *MUC16* and *KMT2D*, but not *TP53* (Figs. 2B–2D). The high mutation burden and frequently mutated genes indicate that at least some endometroid EC tumors belong to "mutation-driven" cancer (*Ciriello et al., 2013*).

To further confirm our observation, we isolated the most significant somatic mutated genes identified in the previous pan-cancer study (*Ciriello et al., 2013*). We further compared the SCNA and somatic mutation frequencies for these genes within these three cancer types (Figs. 2E–2G). As expected, while endometroid EC has substantial number of these genes being more mutated than copy-number changed, the two serous cancer types have dominantly copy-number events rather than mutations ($P < 2.2e-16$, one-side Kolmogorov–Smirnov test).

Taken together, our data analysis suggests that serous-like EC and serous OvCa are similar in the genotypical aberrations of SCNA and SM, whereas endometroid EC stands
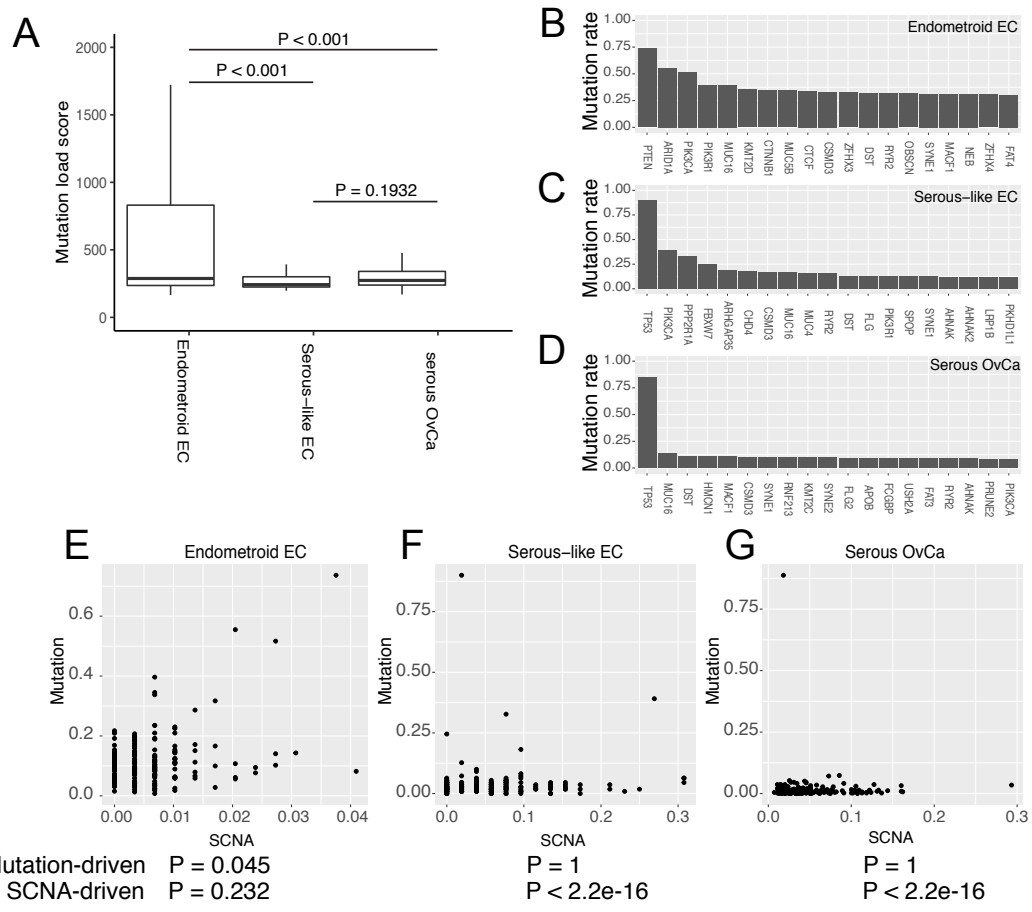

**Figure 2** **The comparison of somatic mutation (SM) among the three cancer types.** (A) The overall SM burdens. P values were resulted from student's *t*-test. (B–D) The top 20 most frequently mutated genes for endometroid EC (B), serous-like EC (C) and serous OvCa (D). The y-axis of the bar plots represents the SM proportions of all available TCGA samples from each indicated cancer type. (E–G) Scatter plots show the rates of gene mutation and gene copy number change for the most important cancer driver genes within each of the three cancer types. *P* values listed below were resulted from Kolmogorov–Smirnovtest comparing the cumulative patterns between gene mutation and gene copy number change.

out as an obvious different cancer type, even though endometroid EC and serous-like EC are originated from the same organ.

## Gene expression and pathway activity

We next checked the difference in gene expression among these three cancer types. Unlike SCNA or SM, Serous-like EC tumors seemed to be in an intermediate status of gene expression between endometroid EC and serous OvCa (Fig. 3A). A more quantitative comparison using sample-wise correlation further showed that serous-like EC has modest similarities to endometroid EC (mean Spearman's correlation 0.270) and serous OvCa (mean Spearman's correlation 0.297), comparing to the low similarity between the latter two (mean Spearman's correlation 0.122) (Fig. 3B). We further speculated that endometroid organ-intrinsic gene expression might be compromised in serous-like EC tumors. To this

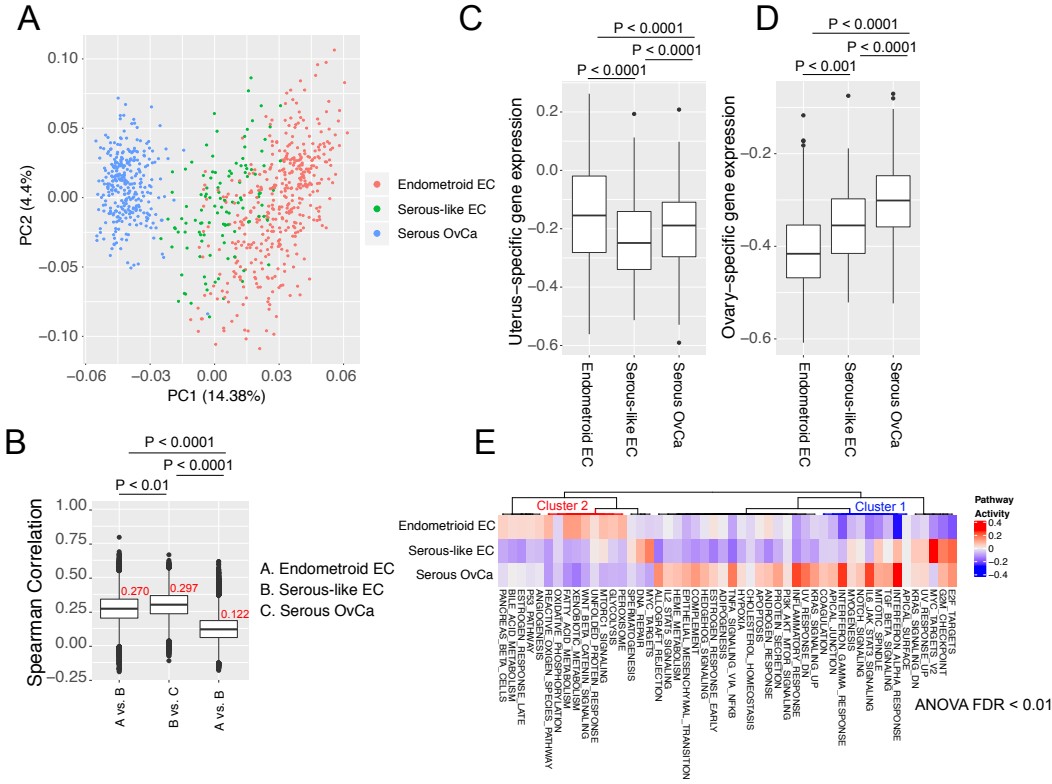

**Figure 3** **The comparison of gene expression and pathway activity among the three cancer types.** (A) PCA plot showing the overall gene expression similarity of all available samples from these three cancer types. (B) Boxplot showing the Spearman's correlation coefficients of all possible sample pairs from different cancer type. $P$ values were resulted from Student's $t$-test. (C–D) Boxplots comparing the organ-specific gene expression, including uterine-specific (C) and ovary-specific (D) from these three cancer types. (E) Relative pathway activities of the hallmark pathways shown by heatmap. For simplicity, only the significantly variable pathways (adjusted $P$ value < 0.01, One-way ANOVA) were shown. Two specific clusters, immune related pathways (cluster 1) and metabolism-related pathways (cluster 2) were highlighted to show the similarities between the two ECs and the two "serous" cancer types, respectively (see main text).

end, we examined the expression of normal uterus-specific (Fig. 3C) and ovary-specific (Fig. 3D) genes in these cancer types. Intriguingly, while endometroid EC and serous OvCa still maintain some organ-specific gene expression profile, serous-like EC seemed to lose the organ-specific gene expression (Figs. 3C–3D).

To gain more biological information, we next summarized gene expression into pathway activities using single sample GSEA (ssGSEA, see method) (*Barbie et al., 2009*) and performed similar comparison. Figure 3E showed the variable hallmark pathways (*Liberzon et al., 2015*) across these three cancer types. Consistent with gene expression, serous-like OvCa has both common pathway activities to endometroid EC and serous OvCa. For instance, some immune response related pathways, such as "inflammatory response", "interferon gamma response", "complement", "IL6-JAK-STAT signaling" are all lower in two ECs than in serous OvCa (cluster 1 in Fig. 3E) (adjusted $p$ value < 0.05,
Student's $t$-test). On the other hand, the serous-like EC tumors also have similar activity for some pathways to serous OvCa, especially those related to metabolism (cluster 2 in Fig. 3E), including "glycolysis", "oxidative phosphorylation", "fatty acid metabolism" and "xenobotic metabolism". Together, serous-like EC displays an intermediate gene expression and signaling pathway activity between the endometroid EC and serous OvCa. In particular, the similarities in pathway activity between the two ECs argues the importance of expression-level examination beyond the clustering based on genetic aberration.

## Survival gene signatures

One unique advantage of cancer data analysis based on TCGA is that the patients' clinical data is available, thus allowing the association between molecular characterizations and survival outcome. We downloaded all the available survival data for these three cancer types and found that the serous OvCa has the worst survival outcome, while the endometroid EC has the best (Fig. 4A). This observation is consistent to our understanding of these three cancer types (*Siegel, Miller & Jemal, 2019*) and suggests that the TCGA survival data are large enough to perform clinically related analyses.

We reasoned that if two cancer types share similar mechanisms for tumor growth, they should also have common survival signature genes, which are related to cancer progression and drug response. To this end, we compared the survival gene signature among these three cancer types. First, we performed univariate Cox proportional-hazards (PH) regression between each gene expression and survival outcome. To summarize the genes with good or bad prognostic values, we extracted the Cox PH coefficients and used them for GSEA analysis to identify pathways that are enriched with these genes. Notably, all three cancer types have their unique signature genes being identified, and these genes are mostly classified into different pathways (Figs. 4B–4D). Nonetheless, the two EC types shared several common good or bad prognostic pathways. For instance, the expression of genes related to "Aminoactyl-RNA biosynthesis" indicates bad prognosis, and immune related genes, including "Allograte rejection" indicate good prognosis for both EC cancer types (GSEA, FDR $< 0.05$) (Figs. 4B–4C). On the other hand, the serous OvCa has unique adverse prognostic pathways, including "Endocytosis", "Focal adhesion", and cell proliferation and growth-related pathways, such as "Ras signaling pathway", "Hippo signaling pathway", "Gastric cancer" and "Glioma" (Fig. 4D). Interestingly, genes involved in "Aminoactyl-RNA biosynthesis" have generally good prognostic values for OvCa, in contrast to the EC cancers (GSEA, FDR $< 0.05$). Our results of "Aminoactyl-RNA biosynthesis" is consistent with a recent report that this pathway activity is only selectively upregulated and linked to tumorigenesis in some cancer types, and one of them is endometrial cancer (*Zhang et al., 2018*).

In terms of detailed survival signature genes, we found that most of them are unique to each of the three cancer types, while the two ECs share several common bad and good prognostic genes (Table S2). For instance, the expression *PHKA1*, one of the phosphorylase kinase regulatory genes (*Pallen, 2003*), has a bad indication of survival outcome for both ECs but not serous OvCa (Figs. 4E–4G) (log-rank test, $p$ value $< 0.05$). Similarly, the expression of *CXCR5*, an important chemokine receptor involved in multiple immune

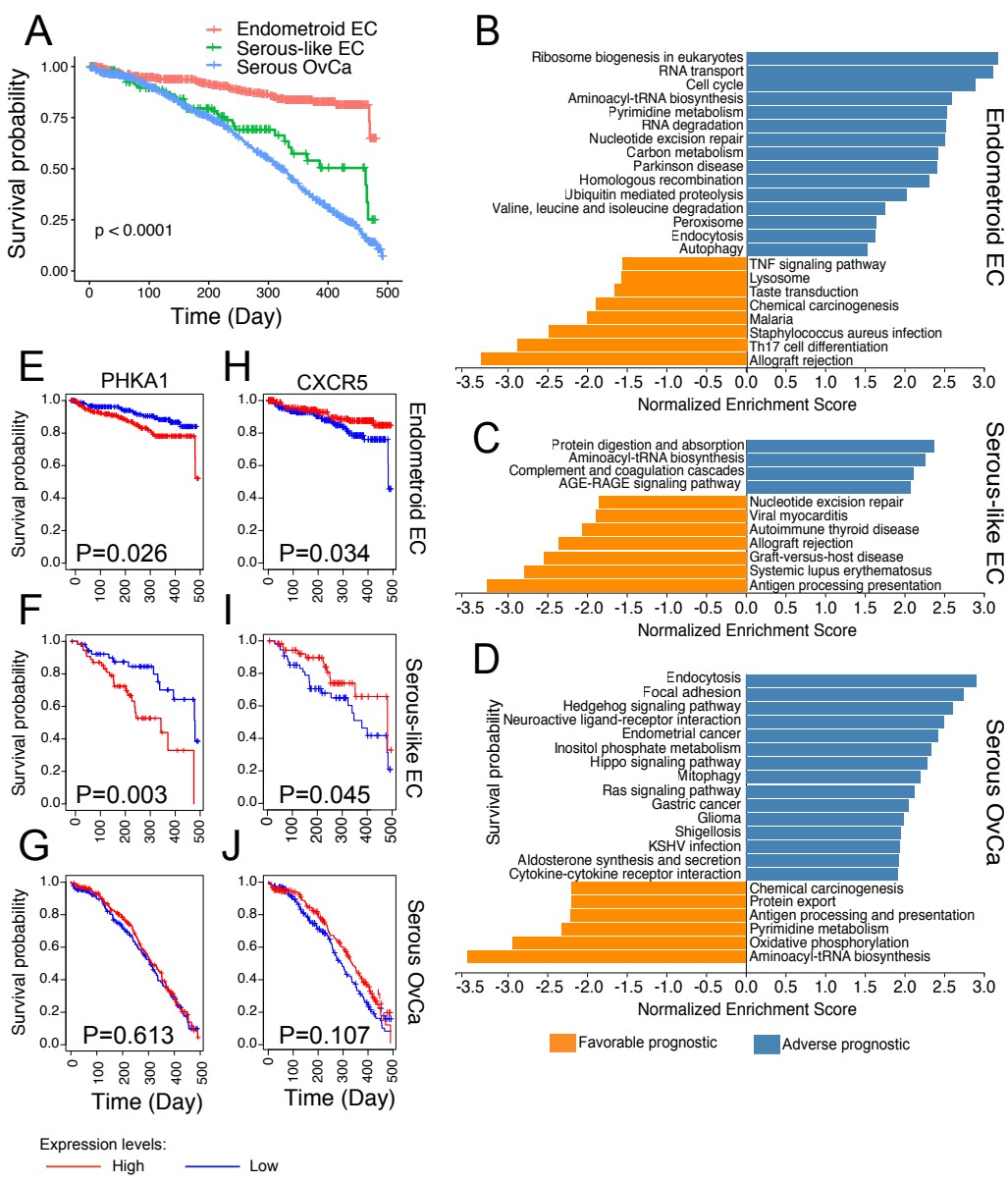

**Figure 4** **Survival signature pathways and genes for the three cancer types.** (A) Kaplan-Meier curve comparing the overall survival for the three cancer types. The P value was resulted from log-rank test. (B–D) Signaling pathways related to survival outcome for endometroid EC (B), serous-like EC (C) and serous OvCa (D). The blue pathways indicate adverse prognostic while the yellow pathways indicate favorable prognostic. All the pathways on display have adjusted P-value less than 0.01 (based on GSEA permutation). (E–J) Kaplan-Meier curve showing two representative genes that have common prognostic values for the two EC types but not serous OvCa. *PHKA1* (E–G) is an adverse prognostic gene while *CXCR5* (H–J) is a favorable prognostic gene.

cell infiltration (*Murphy, 2012*), has a good indication of survival outcome for both ECs (log-rank test, *p* value < 0.05) but not serous OvCa (Figs. 4H–4J). In summary, our survival analysis indicates a closer relationship between the two ECs, although the serous-like EC and serous OvCa share common genetic aberrations.

## Immune infiltration

Finally, we examined the immune infiltration for these three cancer types. Immunotherapy holds great promise for cancer treatment, especially when conventional chemotherapy/radiotherapy and other targeted therapy fail to achieve sufficient response. The efficacy of immunotherapy largely depends on the overall immune infiltration and immune cell composition within the tumor microenvironment (*Alderton & Bordon, 2012*).

We first infer the immune and stromal cell infiltration using RNA expression of previously-established signature genes (*Yoshihara et al., 2013*). Both the serous OvCa and endometroid EC have higher immune infiltration than serous-like EC (Fig. 5A); however, serous OvCa also has high stromal cell infiltration (Fig. 5B, adjusted *p* value < 0.05, Student's *t*-test). Since the stromal cells can contribute to immune suppressive signals (*Valkenburg, De Groot & Pienta, 2018*), there is a possibility that serous OvCa has more immune-suppressive cells than endometroid EC. We thus run CIBERSORT (*Newman et al., 2015*) to explore the detailed immune cell composition within the tumor microenvironment. To our expectation, compared to the other two cancer types, endometroid EC has significantly higher level of cytotoxic immune cells, including CD8 T cells and NK cells, and also Treg cells, whose function is to constrain CD8T cells in tumor (*Mougiakakos et al., 2010*) (adjusted *p* value < 0.05, Student's *t*-test, Fig. 5C). In contrast, the serous OvCa has significant amounts of macrophages and monocytes, which might together form an immune-impressive tumor microenvironment.

Furthermore, we asked whether the cytotoxic CD8 T cells can contribute to the favorable patient survival. Log-rank test analysis found that the higher amount of CD8 T cell is associated with a significantly better survival in endometroid EC (Fig. D) and a better (but not significant) survival in serous-like EC (Fig. E, log-rank test, *p* value < 0.05). In contrast, CD8 T cell infiltration does not show any prognostic value for serous OvCa (Fig. 5F). This might be due to a possibility that the low amount of CD8 T cells is not sufficient to play the anti-tumor role or that there are other immune-suppressive signals to block the function of CD8 T cells.

Together, our results suggest that serous-like EC and serous OvCa have very different immune infiltration profiles. Although serous OvCa and endometroid EC both have high immune infiltration, their immune cell contents are very different from each other and therefore, the potential immunotherapeutic strategy are also likely to be different.

# DISCUSSION

## The multi-omics comparison of three cancer types

Here, we performed detailed analyses based on the TCGA high throughput data to test whether the serous-like EC should be grouped together with serous OvCa into a "serous cancer" type or should stay with the endometroid EC as a typical "endometrial cancer"

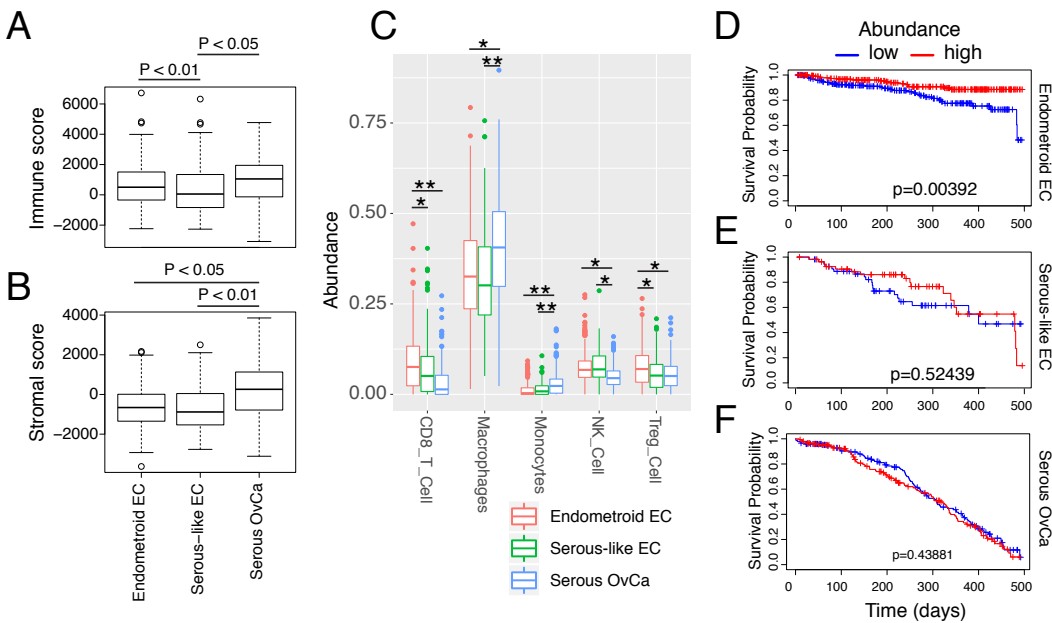

**Figure 5** **Immune infiltration for the three cancer types.** (A–B) Boxplots comparing the overall immune infiltration (A) and stromal infiltration (B) of these three cancer types. Results were calculated using ESTI-MATE. *P* values were calculated by student's *t*-test. (C) Boxplots comparing interested immune cell abundances, including CD8 T cell, Macrophage, Monocyte, NK cell and Treg Cell. Results were calculated using CIBERSORT. *: adjusted *p* value < 0.05; **: adjusted *p* value < 0.01. *P* values were resulted from Student's *t*-test (D–F) Kaplan-Meier curves comparing the prognostic values of CD8 T cell in endometroid EC (D), serous-like EC (E) and serous OvCa (F). *P* values were calculated by log-rank test.

type. Our findings were summarized in Table 2. Although the similarity of SCNA and SM (particularly SCNA) favors the grouping of serous-like EC and serous OvCa, the gene expression and pathway activity, survival gene signature and immune infiltration all point out obvious difference between these two cancer types. Specially, pathway activity and survival gene signature both point to a close relationship between the two ECs than between the two "serous" cancer types. For immune infiltration, the two "serous" cancer types are also very different from each other, in terms of overall immune cell abundance and immune cell composition. Although serous OvCa has high immune cell infiltration, which is similar to endometroid EC, its immune cell composition is largely dominated by immune-suppressive macrophage and monocyte. We noticed that our result about CD8 T cell and serous OvCa was different from a recent clinical study (*Goode et al., 2017*), whose immunohistological analyses showed that high CD8 T cell infiltration favored a better prognosis for serous OvCa. We reasoned that besides cohort difference, the techniques used in these two studies (*in silico* inference based on whole bulk RNA-seq vs. immunohistology focusing on epithelial components of tumor islets) might be detecting CD8 T cells located in different tissue compartments. In particular, the whole-bulk RNA-seq reflects the CD8 T cells distributed across both tumor epithelial and stromal sites. This hypothesis can be further tested by examining the association between stroma-located CD8 T cell infiltration and the clinical outcome. Our results about the discrepancy between the two serous-like

**Table 2 Summarization of all the molecular genotype and phenotype comparisons.** For each molecular event, one or two representative features are shown. The last column of the table indicates the pair of cancer types that is most similar than any other pairs, see the Discussion section for more details.

| Cancer types Molecular events | Endometrioid EC | Serous-like EC | Serous OvCa | Similar pair |
|---|---|---|---|---|
| Copy number alternation | Weak (1q gain) | Strong (Multiple SCNA arms) | Strong (Multiple SCNA arms) | **Serous-like EC and Serous OvCa** |
| Somatic mutation | High (Lead by non-TP53 genes) | Low (Lead by TP53) | Low (Lead by TP53) | **Serous-like EC and Serous OvCa** |
| Overall gene expression and pathway activity | Inflammation low and metabolic high | Inflammation low and metabolic low | Inflammation high and metabolic low | **None** |
| Survival signatures | Aminoacyl-RNA biosynthesis (good) | Aminoacyl-RNA biosynthesis (good) | Aminoacyl-RNA biosynthesis (bad) | **Serous-like EC and Endometroid EC** |
| Immune infiltration profiles | High (cytotoxic) | Low | High (macrophage and monocyte) | **None** |

cancer and the similarity between the two ECs are consistent with a recent report that the tissue-origin largely impacts the cancer type classification (*Hoadley et al., 2018b*).

## Implications for targeted therapy development

Unlike chemotherapy and radiotherapy, the targeted therapies target cancer's specific gene mutations, copy number alterations, proteins, signaling pathways or tumor microenvironment components (*Baudino, 2015*). Detailed molecular profiling and comparative characterizations would be very helpful to delineate tumor groups and develop novel tumor treatment strategies (*Aggarwal, 2010*). For instance, the characterizations on genetic aberration have proven to be important information resources for targeted therapy development, with excellent examples including *BRCA1* mutation, *HER2* amplification and microsatellite instabilities (MSI) (*Tung & Garber, 2018*; *Havel, Chowell & Chan, 2019*; *Oh & Bang, 2019*). In this regarding, serous-like EC and serous OvCa treatment can be benefited from common targeted therapies. Indeed, there have been clinical trials utilizing *EGFR* and *HER2* amplification in these two cancer types (*Wilken et al., 2012*; *Makker et al., 2017*). Based on our analysis, we can also propose that the frequent mutation of *PIK3CA* (Fig. 2B) could be utilized to stratify patients for PIK3CA inhibitor-based treatment. On the other hand, the gene expression, pathways activity and tumor microenvironment characterizations are also too important to neglect. We propose that the substantial differences on these molecular phenotypes are valuable to understand potential responsiveness of targeted therapy and to identify novel therapeutic opportunities. For instance, our observations on immune profiling (Fig. 5) suggest that the immunotherapy should be targeted to inhibit the stromal signals in serous OvCa to first increase the proportion of cytotoxic CD8 T cells or NK cells. In contrast, since there has been a large amount of CD8 T cell in the endometroid cancer, the immunotherapy for this cancer type might be focused on maximizing the function of CD8 T cells using immune checkpoint inhibitors.

## Limitations of this study

Although we strived to perform an unbiased and comprehensive bioinformatic analysis to understand the genetic aberration and molecular phenotypes of endometroid EC, serous-like EC and serous OvCa, we realized that there are several limitations that could not be readily overcome by current datasets and analysis methods. First, the TCGA clinical data do not have detailed records about different therapies (e.g., targeted therapies or clinical trials) each patient received. Therefore, we could not exclude the possibility that some survival results were impacted by the therapeutic difference, instead of intrinsic gene expression. Secondly, there are several molecular phenotypes, such as pathway activity and immune cell infiltration were inferred by *in silico* bioinformatic tools, rather than from experimental tests. There are some controversies and pitfalls in using these tools (*Li, Liu & Liu, 2017*; *Newman et al., 2017*; *Schubert et al., 2018*), although they have all been confirmed by experimental benchmarked when they were originally published (*Hemminki et al., 1998*; *Hanzelmann, Castelo & Guinney, 2013*; *Yoshihara et al., 2013*; *Newman et al., 2017*). Lastly, we want to note that there might be potential batch effects for the gene expression datasets profiled across different cohorts. Although there are several tools designed to adjust batch effect (*Oytam et al., 2016*; *Leek et al., 2018*), it is very difficult to remove batch effect without affecting the true biological signals (*Nygaard, Rødland & Hovig, 2016*; *Bin, Wang & Wong, 2017*; *Newman et al., 2017*). For this consideration, we chose to perform re-normalization across the different cohorts rather than the explicit batch effect correction. Our strategy was similar to several pan-cancer studies (*Rooney et al., 2015*; *Berger et al., 2018*; *Ge et al., 2018*; *Rosario et al., 2018*). We would foresee that with better clinically annotated cohorts, more advanced experimental techniques, such as single-cell multi-omics techniques and more sophisticated data processing methods, these analyses would be largely improved.

## CONCLUSIONS

In summary, our analysis identifies both similarities and discrepancies between serous-like EC and serous OvCa and provides possible clinical contextualization for some of the characterizations. On the genetic profiles, serous-like EC and serous OvCa share very similar SCNA and SM profiles, which was the main reason that they were recently considered to be a uniformed "serous" cancer type. However, there are several important molecular phenotype differences, including gene expression and pathway activity, survival signature genes and immune infiltration. Our analysis indicates that common targeted therapies might be developed to treat serous-like EC and serous OvCa based on mutation drivers, such as *PIK3CA*. Equal amount of considerations, if not more, should be paid to on the gene expression, signal pathway activities and tumor microenvironment to investigate drug responsiveness and to identify novel molecular targets for them individually.

### Funding

The authors received no funding for this work.

## Competing Interests

The authors declare there are no competing interests.

## Author Contributions

- Hui Zhong analyzed the data, conceived and designed the experiments, prepared figures and/or tables, authored or reviewed drafts of the paper, and approved the final draft.
- Huiyu Chen, Huahong Qiu and Chen Huang analyzed the data, prepared figures and/or tables, authored or reviewed drafts of the paper, and approved the final draft.
- Zhihui Wu conceived and designed the experiments, authored or reviewed drafts of the paper, and approved the final draft.

## Data Availability

The code and data matrices for the data analysis are available at GitHub:

https://github.com/ibphuangchen/endometrial_vs_ovarian_comparison.

The original data is available at the GDC data portal (https://portal.gdc.cancer.gov) and using the R package TCGAbiolinks (http://bioconductor.org/packages/release/bioc/html/TCGAbiolinks.html).

For the original data download from GDC or TCGAbiolinks, use the "TCGA-UCEC" and "TCGA-OV" as the project name, "RNA-Seq", "WXS" as the experimental strategy, "normalized results", "Simple somatic mutation" and "Copy Number Variation" for the data type/category, and the TCGA sample barcode listed in the data matrices (accessible via github).

## Supplemental Information

Supplemental information for this article can be found online at http://dx.doi.org/10.7717/peerj.8347#supplemental-information.

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
