# Peer review of "A multiomics comparison between endometrial cancer and serous ovarian cancer"

_PeerJ, doi:10.7717/peerj.8347_

## Round 0.1 · original submission · Major Revisions

The manuscript is clearly asking an important question about endometrial cancer subtypes, but there are some major concerns in study design and implementation. Please review the concerns raised by both reviewers and consider submitting a revised manuscript.

·

Basic reporting

• The study is an interesting topic, and the basic reporting is generally clear.
• Minor grammatical editing is needed
• To emphasize the research questions of interest, specifically state in abstract and intro that you compare the three cancer groups. Initially it seems to be focused on just serous EC and OvCa, but the paper really describes comparisons of the three groups.
• A bit more background on previous findings with TCGA would be helpful. How do these results differ from TCGA PanCancer analyses? Was EC included in those papers? Have pathway analyses previously been performed on this data?

Experimental design

• More detail should be provided on imputation of gene expression. How much data was missing?
• Need more detail on how the mutation burden was calculated. Was it a sum of the mutations across all genes? Can you provide references if standard methods were used?
• Provide some motivation for the use of single sample GSEA in the pathway analyses, as opposed to the traditional GSEA method used in the survival analysis.
• How did you deal with potential batch effects in the RNAseq data? Presumably the two types of EC may have been sequenced at the same time, but OvCa was presumably done completely separately. This would confound any across cancer type comparisons, and has direct implications for the validity of the results below.

Validity of the findings

• Many of the results seem very descriptive, and statements made comparing EC and OvCa are often made without formally testing or comparing the groups, such as the results displayed in Figure 1A and B. In Figure 1B, if the serous-like EC points would be plotted last, it would be much easier to see how they group compared to the larger sample size endo EC and serouc OC types; the smaller number of serous EC points are hidden.
• Again, the mutation burden results description of copy-number driven vs. mutation-driven cancers would be better supported with formal tests.
• Line 181 in the pathway results says that the two ECs share common gene expression related to tumor progression, but this is not supported by Figure 3B, which shows different patterns.
• Line 185-186: The statement ‘Consistent with gene expression, the two ECs share more common pathway profiles between each other than to the OvCa’ doesn’t seem to be supported by the figure, where the expression of all types seem distinct. Can you show a formal comparison of this?
• Line 184 incorrectly refers to Figure 3B (should be Figure 3C). This figure needs a more descriptive legend of the heatmap. What is the dendogram representing?
• In Figure 4 and the immune comparisons, were any direct statistical comparisons made or are all results descriptive? It seems that you should be comparing the hazard ratios across the cancer types.
• Figure 4D: the survival relationship of DCTN6 is not similar between the two ECs, they go in opposite directions!
• OvCa CD8 and survival associations disagree with prior reports of a strong association between higher CD8 TILs and better survival in high-grade serous ovarian cancer (Goode et al, JAMA Oncology, 2017). This conflicts with what is shown in Figure 5C.
• In the immune comparisons, were any direct statistical comparisons made or are all results descriptive? This isn’t clear in the methods section. Line 244 describes significantly higher levels for EC, but compared to what?
• In general, the conclusions are over-generalized.

Additional comments

• In general, the manuscript poses some interesting scientific questions, asking whether serous EC is molecularly more similar to endometrioid EC or serous OC.
• Consider presenting copy number and mutation results under separate headings, since they are measuring different things
• For survival associations, did you consider the fact that these cancers are treated differently (as alluded to in the text)?
• Limitations to the study should be outlined.
• Table 2 is helpful to summarize the findings

Reviewer 2 ·

Basic reporting

- The manuscript uses, most of the time, clear and unambiguous, professional English, however, I would suggest to homogenize the use of acronyms throughout the paper, e.g., OvCa and OV are used in the same paragraph in the abstract to refer to Ovarian Carcinoma (Lines 17 and 19).

- If the main topic of the manuscript is the use of similar/different therapies for serous-like endometrial carcinoma and serous ovarian cancer, I would like to see more background/context about the current therapies for both cancer types. There is a relevant review which should be cited (Brasseur, K., Gévry, N. & Asselin, E. Chemoresistance and targeted therapies in ovarian and endometrial cancers. Oncotarget 8, 4008–4042 (2016)).

- The figures are clear, however, some of them are not vectorized figures. The authors shared their scripts in a GitHub repository, however, they did not share the raw data to run them and/or README files to be able to reproduce the figures.

- The authors were able to use TCGA data to compare the global somatic signatures and gene expression and immune infiltration profiles of serous-like endometrial cancer and ovarian cancer, however, these results are not conclusive to determine whether or not these two cancer types could share therapeutic targets.

Experimental design

- The manuscript contains a well structured methods section, however, the methods are not described with sufficient detail & information to replicate.

- In Fig 3B, the authors mention a set of uterus-specific and ovary-specific genes, however, I did not see them list in the manuscript. How many genes are in each set?

Validity of the findings

- The authors use TCGA data (clinical data, RNA expression, somatic copy number alterations and somatic mutation) to compare endometrial and ovarian cancers. The authors showed serous-like EC and serous OvCa have similar somatic CNA and somatic mutation signatures, but different global gene expression, survival signatures and immune infiltration profiles. They suggest endometrioid and serous-like EC "share several common bad and good prognostic genes", however, there did not perform a test to determine the statistical significance of the number of "common bad and good prognostic genes" and compare that to OvCa; also, the authors used DCTN6 as an example of "good indication of survival outcome for both ECs", but Fig 4D shows they have opposite prognostic effects in the two EC subtypes. I suggest a more strict approach when comparing the survival signatures between these three cancer subtypes.

- Using this molecular characterization, they concluded "against treating serous-like EC and serous ovarian OvCa as a simple "serous" cancer type". All the results combined showed similarities (CNV and somatic mutations) and differences (gene expression and immune infiltration) between serous-like EC and serous ovarian cancer, however, the authors did not address why the similarities found in these cancer types, are not important to be considered to have similar therapeutic targets.

Additional comments

Overall, the molecular characterization performed to compare serous-like endometrial cancer and serous ovarian cancer requires a more robust statistical framework.

The conclusion of arguing "against treating serous-like EC and serous ovarian OvCa as a simple "serous" cancer type" is not completely supported by the results, specifically the similarity of the somatic CNA and mutations signatures.

---

## Round 0.2 · Minor Revisions

Thank you for submitting a revised manuscript. Please see the reviewer responses to your revisions, particularly the comments about batch effects and typographical issues.

·

Basic reporting

For the most part, my comments were addressed. There are still some minor grammatical issues. For example, in the introduction line 45 of the pdf 'income' should be 'outcome',
and line 75 'ratiotherapy' should be 'radiotherapy'. I noticed some issues with the references. The survival R package was written by Terry Therneau (one person), not Terry and Therneau (two people), and the added reference in the discussion for the study of CD8 and survival and ovarian cancer (line 330) should be Goode et al, JAMA Oncology, not Leiber et al.

Experimental design

My previous comments were addressed. Although I believe some text regarding the possible batch effects should be added to the discussion in the limitations section.

Validity of the findings

My comments were adequately addressed, although I think some refinement is needed for discussion of the conflicting results regarding the CD8 survival association in OVCA. The Goode paper did not target specific types of CD8 cells, but total CD8+ T cells, as was done here. Rather, the difference in findings is likely due more to the first point the authors make regarding the tumor heterogeneity. The analyses we conducted in the Goode paper were restricted to epithelial cells, while the mRNA assessed here in TCGA would included stroma. Differences could likely be due to stromal infiltration. In fact, when we look at CD8 gene expression from Nanostring data that was generated from a subset of the same subjects as assessed in the Goode et al paper, the association between CD8 and survival is much weaker. I think differences with TCGA could also be a power issue, as the Goode paper include close to 5000 cases, where the TCGA analysis is only around 300.

Additional comments

No further comments

Reviewer 2 ·

Basic reporting

The authors corrected homogenized most of the acronyms throughout the paper, but I think they missed some OvCa/OV in the abstract.

The authors have complemented the introduction/background with additional references relevant for providing the right context for this study.

The authors mentioned that they have uploaded the data matrices they used in this project to their GitHub repository, which is a very valuable aspect to reporting a study. They also included extra figures, and I recommend the authors make some changes to make sure all figures are large enough to visualize when printed, e.g., the boxplots in Figure 1C are too narrow and can get distorted when printed.

The authors have made changes that are beneficial for the understanding of the relevance of the results.

Experimental design

The authors have updated the methods section to "rephrase ambiguities" making the methods section clearer and more easy to replicate.

The authors provided additional supplementary information for a more clear understanding of all the datasets used.

Validity of the findings

The authors have updated the results/discussion sections to address discrepancies between the main conclusions and the results selected to support the conclusions. The authors now provide good evidence for the conclusions.

Additional comments

Bases on the previous revision, the authors have performed a more robust statistical comparison between the two cancer subtypes.

The authors have stressed the importance of showing the differences between the two cancer subtypes, which I agree, but now have rephrase the conclusions to make the more precise.

---

## Round 0.3 · accepted · Accept

Thank you very much for the submission of a revised version of your paper. I have gone through the revised, track-changes manuscript and rebuttal letter, and see that the authors addressed the reviewers' concerns. So, based on my own assessment as an editor, no further revisions are required, and the manuscript may be now accepted for publication in its current form.